# Well-Differentiated Papillary Mesothelioma of the Peritoneum Is Genetically Distinct from Malignant Mesothelioma

**DOI:** 10.3390/cancers12061568

**Published:** 2020-06-13

**Authors:** Raunak Shrestha, Noushin Nabavi, Stanislav Volik, Shawn Anderson, Anne Haegert, Brian McConeghy, Funda Sar, Sonal Brahmbhatt, Robert Bell, Stephane Le Bihan, Yuzhuo Wang, Colin Collins, Andrew Churg

**Affiliations:** 1Vancouver Prostate Centre, Vancouver, BC V6H 3ZH, Canada; raunakman.shrestha@ucsf.edu (R.S.); nabavinoushin@gmail.com (N.N.); svolik@prostatecentre.com (S.V.); sanderson@prostatecentre.com (S.A.); ahaegert@prostatecentre.com (A.H.); brian.mcconeghy@ubc.ca (B.M.); fsar@prostatecentre.com (F.S.); sonal.brahmbhatt2001@gmail.com (S.B.); rbell@prostatecentre.com (R.B.); slebihan@prostatecentre.com (S.L.B.); ywang@bccrc.ca (Y.W.); 2Department of Urologic Sciences, University of British Columbia, Vancouver, BC V5Z 1M9, Canada; 3Department of Radiation Oncology, University of California San Francisco, San Francisco, CA94143, USA; 4Department of Experimental Therapeutics, BC Cancer Agency, Vancouver, BC V5Z 1L3, Canada; 5Department of Pathology, Vancouver General Hospital, Vancouver, BC V5Z 1M9, Canada

**Keywords:** well-differentiated papillary mesothelioma, WDPM, malignant mesothelioma, DNA sequencing, mutation

## Abstract

Well-differentiated papillary mesothelioma (WDPM) is an uncommon mesothelial proliferation that is most commonly encountered as an incidental finding in the peritoneal cavity. There is controversy in the literature about whether WDPM is a neoplasm or a reactive process and, if neoplastic, whether it is a variant or precursor of epithelial malignant mesothelioma or is a different entity. Using whole exome sequencing of five WDPMs of the peritoneum, we have identified distinct mutations in *EHD1*, *ATM*, *FBXO10*, *SH2D2A*, *CDH5*, *MAGED1*, and *TP73* shared by WDPM cases but not reported in malignant mesotheliomas. Furthermore, we show that WDPM is strongly enriched with C > A transversion substitution mutations, a pattern that is also not found in malignant mesotheliomas. The WDPMs lacked the alterations involving *BAP1*, *SETD2*, *NF2*, *CDKN2A/B*, *LASTS1/2*, *PBRM1*, and *SMARCC1* that are frequently found in malignant mesotheliomas. We conclude that WDPMs are neoplasms that are genetically distinct from malignant mesotheliomas and, based on observed mutations, do not appear to be precursors of malignant mesotheliomas.

## 1. Introduction

Well-differentiated papillary mesothelioma (WDPM) is a morphologically distinctive papillary proliferation of mesothelial cells that is most commonly encountered as an incidental finding in the peritoneal cavity, and less often in the pleural cavity, pericardium, and tunica vaginalis. These lesions may be single or multiple but by definition do not invade the underlying stroma and usually behave in a benign or indolent fashion, sometimes persisting for many years [1]. However, the nature of WDPM is disputed, with theories ranging from a reactive non-neoplastic process to a benign tumor, to a variant and/or precursor of epithelial malignant mesotheliomas [2]. To add further confusion, unequivocal invasive malignant mesotheliomas can have areas that mimic WDPM. Since malignant mesotheliomas are aggressive tumors, the distinction from WDPM is important, but WDPMs are sometimes treated with debulking cytoreductive surgery followed by hyperthermic intraperitoneal chemotherapy (HIPEC) as if they were mesotheliomas [3].

Genome-wide sequencing analyses of malignant mesotheliomas have revealed frequently observed genomic aberrations such as loss of function mutation and/or copy number alterations/deletion of *BAP1*, *SETD2*, *CDKN2A*, and *NF2* [4,5,6]. Studies analyzing WDPM using DNA sequencing technology are limited. Case studies have reported WDPMs with somatic mutation of *E2F1* [7], heterozygous loss of *NF2* [8], and germline *BAP1* mutation [9], which if correct would suggest that they may be variants of malignant mesothelioma. Nevertheless, using immunohistochemistry (IHC) and fluorescence in situ hybridization (FISH), Lee et al. demonstrated that, unlike in malignant mesothelioma, both *BAP1* and *CDKN2A* are intact and respective proteins are expressed in WDPMs [10]. More recently, Stevers et al. [11] performed genomic profiling of 10 WDPMs and found that they harbored *TRAF7* or *CDC42* mutually exclusive missense mutations.

To shed further light on this question we performed an extensive genomic characterization of a cohort of five WDPMs of the peritoneum.

## 2. Results

### 2.1. Histopathological Features of WDPM

We assembled a cohort of five incidentally identified WDPM cases in the peritoneum detected during surgery for another process and all were solitary lesions. All of these five cases had the typical features described for WDPM [12], i.e., a papillary architecture with a single layer of covering bland mesothelial cells and myxoid cores in the papillae (Figure 1).

### 2.2. Mutational Landscape of WDPM

We performed high-coverage whole exome sequencing of five WDPMs from formalin-fixed and paraffin embedded (FFPE) samples. We achieved a mean sequencing reads coverage of 87×–117×, with at least 20–45% of targeted bases having a coverage of 100× (Appendix A). Due to papillary architecture, the tumor cellularity of the WDPM tissues was estimated to be about 50% (Appendix A). Although the high coverage sequencing provides us an opportunity to detect higher proportions of mutations, the normal tissue admixture lowers the mutation detection sensitivity. To overcome this challenge, we implemented strict mutation filtering criteria as described in the Methods section and retained only high confident mutation calls for downstream analysis.

Analysis of the mutational patterns in WDPM revealed a strong enrichment of C > A transversion substitution mutation (Figure 2A). Using the software deconstructSigs [13], we evaluated the characteristic mutation patterns in WDPM against the mutational signature obtained from the COSMIC mutational signature database [14]. Intriguingly, we identified consistent patterns of nucleotide substitution mutation associated with WDPM. Notably, we found that mutational signature 24 is significantly operative in all five WDPM cases (Figure 2B). In addition to this, mutational signature 21 and 28 were also observed in the WDPM cases.

We identified 461 unique non-silent mutations across five WDPM samples affecting 297 unique protein coding genes (Appendix A). Patient WDPM-04 had the highest mutation burden and WDPM-01 had the least. Two genes—*FBXO10* and *SH2D2A*—were mutated in all five WDPM cases, again displaying consistent mutational patterns (Figure 2C). Missense mutation *EHD1*^D147A^ in the dynamin protein domain was found in four cases (Figure 2C,D). The variant allele frequency (VAF) of *EHD1* was in the range 29–43%, indicating its likely clonal origin (given that the tumor cellularity of the WDPM tissues were estimated to be about 50%) (Appendix A). Notably, we identified missense mutation in DNA-damage response gene *ATM* in four cases (Figure 2C,E). All four cases harbored *ATM*^K2303R^ located in the FRAP-ATM-TRRAP (FAT) domain in the ATM protein. The VAF of *ATM* was also in the range 25–30%, indicating its likely clonal origin (Appendix A). The gene encoding cadherin 5 (*CDH5*) harbored *CDH5*^D714E^ mutations in its C-terminus cadherin protein domain in four cases (Figure 2C,F). The VAF of *CDH5* was also in the range 26–38%, indicating its likely clonal origin (Appendix A). We also identified missense mutation *FBXO10*^C42F^ in four cases and *FBXO10*^C26F^ in one case (Figure 2C,G). Both mutation variants of *FBXO10* were present in the F-box like protein domain. The VAF of *FBXO10* was also in the range 24–37%, indicating its likely clonal origin (Appendix A). Similarly, we identified missense mutation *SH2D2A*^G155V^ in four cases and *SH2D2A*^G155D^ in one case (Figure 2C,H). These variants were located in the SH2 protein domain. The VAF of *SH2D2A* in WDPM-04 was 69%, indicating the mutation to be clonal. The VAF of *SH2D2A* in the rest of the four WDPMs was in the range 37–47% (Appendix A). Furthermore, we also identified mutations in *MAGED1* and *TP73* in each of the four WDPM cases (Figure 2C).

### 2.3. Copy Number Landscape of WDPM

The aggregate copy number aberration (CNA) profile of WDPM is shown in Appendix A. We observed 278 CNA events across all samples (Appendix A). The CNA resulted in alterations of about 4–14% of the protein-coding genomes in the WDPM. Patient WDPM-02 had a high copy number burden and WDPM-03 had the least copy number burden (Appendix A). Overall, copy number profiles of the WDPM did not show many alterations (Appendix A). Notably, we found copy number gain of *SETDB2* and *LAST2* and copy number loss of *SMARCA4* and *TRAF7* in WDPM-02. We also found copy number loss of cancer genes such as *CCNE1*, *MAF*, *MAFB*, *MYC*, *ZNF479*, and *MGMT* and copy number gain of *FOXA2*, *CDH10*, and *GPC5* in at least two WDPM cases.

### 2.4. Signaling Pathways Dysregulated in WDPM

To identify signaling pathways dysregulated by mutated genes in WDPM, we performed pathway enrichment analysis using the KEGG [15] pathway database (see Methods section). Our analysis revealed that WDPM mutations target different signaling pathways often dysregulated in cancer (Figure 3 and Appendix A) such as pathways in cancer, focal adhesion, Vascular endothelial growth factor (VEGF) signaling, Janus kinases - signal transducer and activator of transcription (JAK-STAT) signaling, Wnt signaling, P53 signaling, apoptosis, etc. We found *CDH5* mutations target cell adhesion and the leukocyte migration pathway, *EHD1* mutations target endocytosis, *SH2D2A* mutations target the VEGF signaling pathway, *ATM* mutations target apoptosis and P53 signaling pathways, and *TP73* targets the neurotrophin signaling and P53 signaling pathways. This indicates that the mutations identified in WDPM cases might be relevant to pathogenesis of WDPM.

### 2.5. WDPM is Genetically Distinct from Malignant Mesothelioma

Next, we compared the genomic profiles of WDPM with those of malignant peritoneal mesothelioma. For this, we leveraged the DNA sequencing data from two recently published peritoneal mesothelioma patient cohorts [6,16]. We first assessed the pattern of mutations in WDPM and peritoneal mesothelioma cases. Intriguingly, we observed that WDPM has a strong enrichment of C > A transversion substitution mutation (Figure 2A,B), whereas, peritoneal mesothelioma has strong enrichment of C > T transition substitution mutation (Appendix A). This mutational pattern in WDPM is different from those reported in pleural [4,5] or peritoneal [6] mesotheliomas.

Notably, we found WDPM specific mutations in *EHD1*, *FBXO10*, *CHD5*, *MAGED1*, *ATM*, and *TP73* genes that were absent in peritoneal mesothelioma (Figure 4A). Although mutations in *EHD1* and *ATM* genes were each observed in peritoneal mesothelioma, we did not find the WDPM-specific *EHD1*^D147A^, *EHD1*^A465D^, and *ATM*^K2303R^ mutations in these cases. Interestingly, in WDPM, we did not find any of the mutations in *BAP1*, *SETD2*, *TP53*, *NF2*, *CDKN2A*, and *LAST1/2* frequently observed in malignant mesotheliomas (Figure 4A). We also did not find mutations in *TRAF7* or *CDC42* in WDPM, however, *TRAF7* mutations were observed in several peritoneal mesothelioma cases. Furthermore, we evaluated the differences in the copy number status of genes between WDPM and peritoneal mesothelioma. We did not find any copy number loss in gene characteristics of malignant mesotheliomas such as *BAP1*, *SETD2*, *PBRM1*, *SMARCC1*, *CDKN2A/B*, *LATS1/2*, and *NF2* (Figure 4B). *TRAF* copy number loss was observed in one WDPM case, whereas, several peritoneal mesothelioma cases harbored *TRAF7* copy number alteration.

## 3. Discussion

In this study, we investigated the genomic alterations found in a cohort of five WDPMs. The tumors analyzed here are clinically typical of the setting in which WDPM is most commonly found, i.e., as an incidental lesion discovered during surgery for another process, and all lesions were morphologically characteristic WDPM.

Overall, our results suggest that WDPM are distinctive lesions with their own set of genomic alterations. Given the number of mutations and the nature of the mutations found, including at least one tumor suppressor gene, *TP73*, and several genes that may be associated with other types of malignancy (*ATM*, *CDH5*, *MAGED1*) [17,18,19], WDPM clearly appears to be a functionally benign neoplasm and not a reactive process. Further, it is clear that WDPM are genetically quite different from both peritoneal and pleural mesotheliomas. Indeed, our most important finding is the lack of alterations involving *BAP1*, *SETD2*, *NF2*, *CDKN2A*, *PBRM1*, and *SMARCC1* genes consistently mutated or deleted in malignant mesotheliomas.

We found consistent mutation patterns in five WDPMs with strong enrichment of C > A transversion substitution mutation and COSMIC mutational signature 24. The WDPMs harbored distinct mutations in *EHD1*, *FBXO10*, *CHD5*, *MAGED1*, *ATM*, and *TP73* genes either in all five or at least four out of five WDPM cases. The COSMIC mutational signature 24 has been shown to be commonly found in certain liver cancers with exposure to carcinogen such as aflatoxin [20]. However, these WDPMs were incidental findings during surgery and any prior exposure to carcinogens (either aflatoxin or asbestos) is extremely unlikely. Mutations and copy number changes in *CDH5* have been previously reported in mesotheliomas [21,22] but are uncommon events and were not present in any of our reference mesothelioma datasets (Figure 3). *CHD5* is known to promote intravasation and stimulates TGF-β driven epithelial–mesenchymal transition (EMT) [23]. *EHD1* regulates the endocytic recycling process. *EHD1* is known to play a key role in transportation of receptors from endosomes into the endocytic recycling compartment (ERC) and from the ERC to the plasma membrane [24]. Moreover, *EHD1* has been associated with cell proliferation, apoptosis, metastasis, and drug resistance in breast and non-small cell lung cancer (NSCLC) [25] but has not been reported to be abnormal in malignant mesotheliomas. *FBXO10* binds to the anti-apoptotic oncoprotein BCL-2 and promotes its degradation, thereby initiating cell death in lymphomas [26]. *SH2D2A* is known to be involved in T-cell activation [27]. Mutations in *FBXO10*, *SH2D2A*, and *TP73* has not been reported in any malignant mesotheliomas.

Our study confirms a lack of copy number alterations in *BAP1*, *SETD2*, *PBRM1*, *SMARCC1*, *CDKN2A/B*, *LATS1/2*, and *NF2*. Copy number loss of *BAP1*, *SETD2*, *PBRM1*, and *SMARCC1* is often observed in peritoneal mesothelioma [6,28]. Copy number loss of *BAP1*, *CDKN2A/B*, *LAST1/2*, and *NF2* is frequently found in pleural mesothelioma [4,5].

What is surprising in our results is the absence of the TRAF7 and CDC42 alterations reported by Yu et al. [7] and Stevers et al. [11] in WDPM and by the same group in adenomatoid tumors [29]. Alterations in TRAF7 have also been reported in malignant mesotheliomas [4,16,30]. However, this does not appear to be a case of tumor misclassification, since the lesion illustrated by Stevers et al. [11] is a very typical WDPM and is identical to the tumors analyzed here. The lesions analyzed by Stevers et al. [11] were also all incidental findings and 8/10 were solitary, as were ours, and the lesions for which they had follow up did not behave in a malignant fashion.

The exact reasons for the discrepancy between our study and those of Stevers et al. [11] are unclear. It is possible that the underlying populations are genetically different, particularly given the very large and diverse immigrant population in Vancouver, Canada. The analytical approach used in these two studies was also somewhat different. Stevers et al. [11] used a targeted panel consisting of 479 cancer-related genes (UCSF500 Cancer Panel) for sequencing (Illumina HiSeq 2500), whereas we used Ion AmpliSeq™ (Thermo Fisher Scientific, Waltham, MA, USA) exome sequencing which covers 18,961 genes (Ion Proton™). The overlap in the genes examined between these two studies is given in Appendix A. Using a targeted panel provided Stevers et al. [11] an advantage to sequence a small number of genes at a high depth (average depth = 320×, range = 33×–722×), whereas we sequenced a large number of genes at a cost of sequencing depth (average depth = 102×). Stevers et al. [11] reported 21 mutations covering 10 genes in 10 WDPM cases, whereas we identified 461 mutations covering 297 genes in 5 WDPM cases. There is no overlap of the mutated genes reported in Stevers et al. [11] and our study (Appendix A). In fact, the UCSF500 gene panel used by Stevers et al. [11] covered only 10 mutated genes reported by our study (Appendix A). We note that, despite high sequencing depth, no mutations in *ATM* (which was examined in the UCSF500 panel) were reported by Stevers et al. [11], whereas we identified consistent *ATM*^K2303R^ mutations in 4 out of 5 WDPM cases (Appendix A). We did identify a few low confidence *TRAF7* mutations, but these did not pass our mutation filtering criteria (see Appendix A and Appendix B for detailed information). These differences likely indicate genomic heterogeneity in WDPM and warrants further investigation in larger patient cohort settings. Once there are sufficient cases described with consistent results, it may be possible to use a genomic approach to decide whether an equivocal case is a WDPM or a malignant mesothelioma and to base treatment on such data.

## 4. Materials and Methods

### 4.1. Patient Cohort Description and Tissue Procurement

A cohort of incidentally identified WDPM tissues (n = 5) were assembled from the surgical pathology archives at the Vancouver General Hospital. This study was approved by the Institutional Review Board of the University of British Columbia and the Vancouver Coastal Health (REB No. H15-00902 and V15-00902).

### 4.2. Whole Exome Sequencing

DNA from marked FFPE tissue sections (5–10 µm in thickness, ~50% WDPM cellularity) were isolated using a truXTRAC FFPE DNA Kit with Covaris Adaptive Focused Acoustics^®^ (AFA^®^) technology, which enables the removal of the paraffin from the FFPE tissue in SDS buffer while simultaneously rehydrating the tissue. The samples were treated with proteinase K 0.2 mg/mL (Roche) followed by overnight incubation at 55 °C. After post-incubation in proteinase K, the samples were treated with RNAse and DNA extracted as per the truXTRAC FFPE DNA extraction protocol (cat#: 520136, Covaris, Inc., Woburn, MA, USA). The amount of DNA was quantified using Qubit^®^ dsDNA HS Assay (Thermo Fisher Scientific).

For Ion AmpliSeq™ (Thermo Fisher Scientific) exome sequencing, 100 ng of DNA was used as input for Ion AmpliSeq™ Exome RDY library preparation, a PCR-based sequencing approach using 294,000 primer pairs (amplicon size range 225–275 bp), which covers >97% of consensus coding sequence (CCDS) (Release 12), >19,000 coding genes, and >198,000 coding exons. Libraries were prepared, quantified using qPCR, and sequenced according to the manufacturer’s instructions (Thermo Fisher Scientific). Samples were sequenced on the Ion Proton System using the Ion PI™ Hi-Q™ Sequencing 200 Kit and Ion PI™ v3 chip. Two libraries were run per chip for a projected minimum coverage of 40 million reads per sample.

### 4.3. Single Nucleotide Variant Calling

We used Torrent Server (Thermo Fisher Scientific) for mapping aligned reads to the human reference genome hg19 (Torrent Mapping Alignment feature). Variants were identified using a Torrent Variant Caller plugin with the optimized parameters for AmpliSeq exome sequencing (Thermo Fisher Scientific). The variant call format (VCF) files from all samples were annotated using ANNOVAR [31].

To account for the low tumor cellularity in the WDPM samples and the absence of the matched control samples, we used strict mutation calls filtering criteria. Mutations were retained if (a) allele frequency (AF) < 75%, (b) read quality pass > 50%, (c) average heterozygosity < 0.1, (d) mutation calls not present in dbSNP database. We filtered out all In-Dels from our variant calls. Non-silent exonic variants including non-synonymous single nucleotide variations (SNVs), stop-codon gain SNVs, stop-codon loss SNVs, splice site SNVs, and frameshift In-Dels in coding regions were retained if they were supported by more than 50 reads. Furthermore, putative variants were manually scrutinized on the Binary Alignment Map (BAM) files through Integrative Genomics Viewer (IGV) version 2.3.25 [32]. Furthermore, due to lack of matched germline control samples from the WDPM cases, we used genomic DNA samples from blood of a cohort of peritoneal mesothelioma patients as germline control samples. We filtered out any variants that were also present in these control samples [6]. In this way, we excluded any potential germline variants as well as false positive calls and obtained highly confident variants of WDPM. Based on the variant allele frequency (VAF), the mutations identified in WDPM were clustered into different groups using the R-package Maftools [33].

### 4.4. Copy Number Aberration (CNA) Calls

Copy number changes were assessed using Nexus Copy Number Discovery Edition Version 8.0 (BioDiscovery, Inc., El Segundo, CA, USA). Nexus NGS functionality with the FASST2 Segmentation algorithm was used to make copy number calls (a circular binary segmentation/hidden Markov model approach). The significance threshold for segmentation was 5 × 10^−6^ with a minimum of 3 probes per segment and a maximum probe spacing of 1000 between adjacent probes before breaking a segment. The log ratio thresholds for single copy gain and single copy loss were set at +0.2 and −0.2, respectively. The log ratio thresholds for homozygous gain/loss were set at +0.6 and −1.0, respectively. The tumor BAM files were processed and compared with BAM files from a normal tissue pool as reference control. Reference reads per CN point (window size) was set to 8000. We used the Genomic Identification of Significant Targets in Cancer (GISTIC) [34] algorithm in Nexus to identify significantly amplified or deleted regions across the genome. The amplitude of each aberration is assigned a G-score as well as a frequency of occurrence for multiple samples. The false discovery rate (FDR) q-value for the aberrant regions was set to a threshold of 0.15.

### 4.5. Mutational Signature Analysis

We used deconstructSigs [13] software, a multiple regression approach to statistically quantify the contribution of mutational signatures for each tumor. The 30 mutational signatures were obtained from the COSMIC mutational signature database [14]. Only non-silent mutations were used to obtain the mutational signatures. In brief, deconstructSigs attempts to recreate the mutational pattern using the trinucleotide mutation context from the input sample that closely resembles each of the 30 mutational signatures from the COSMIC mutational signature database. In this process, each mutational signature is assigned a weight normalized between 0 to 1 indicating its contribution. Only those mutational signatures with a weight more than 0.06 were considered for analysis.

### 4.6. Pathway Enrichment Analysis

The mutated genes were tested for enrichment against signaling pathways present in the KEGG [15] pathway database obtained from the Molecular Signature Database (MSigDB) v6.0 [35]. A hypergeometric test-based gene set enrichment analysis was used for this purpose (https://github.com/raunakms/GSEAFisher). A cut-off threshold of Benjamini–Hochberg (BH) corrected *p*-value < 0.01 was used to obtain the significantly enriched pathways. Only pathways that are enriched with at least three mutated genes were considered for further analysis.

### 4.7. Peritoneal Mesothelioma Datasets

We utilized DNA sequencing datasets of two publicly available patient cohorts of peritoneal mesothelioma—VPC cohort [6] and AACR Project GENIE Cohort [16]. We used mutation and copy number profiles from both datasets for comparison with the genomic profiles of WDPM cases. AACR GENIE project data, Version 5.0, were downloaded from https://www.synapse.org/#!Synapse:syn7222066.

## 5. Conclusions

We have shown that WDPM are genetically distinct from malignant mesotheliomas and in our hands have a characteristic pattern of C > A transversion substitution mutations; *EHD1*, *FBXO10*, *CHD5*, *MAGED1*, *ATM*, and *TP73* missense mutations; as well as enrichment of COSMIC mutation signature 24. Taken in conjunction with the data from Stevers et al., these findings further reinforce the idea that WDPM should not be treated in the same fashion as malignant mesotheliomas.

## Figures and Tables

**Figure 1 cancers-12-01568-f001:**
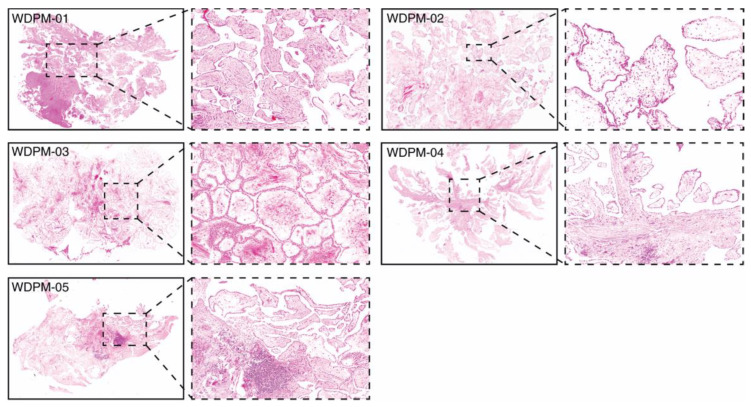
Histopathology of five WDPM cases used for the study. Microphotographs of histological features of WDPM stained using haematoxylin and eosin (H&E). The panel under the dotted box represents the magnified section of the photomicrographs at ×20. The lesion sites/sizes were peritoneum, site not specified, for cases WDPM-01 (3 mm), WDPM-02 (6 mm), WDPM-03 (4 mm), WDPM-04 mesentery (4 mm), and WDPM-05 omentum (4 mm).

**Figure 2 cancers-12-01568-f002:**
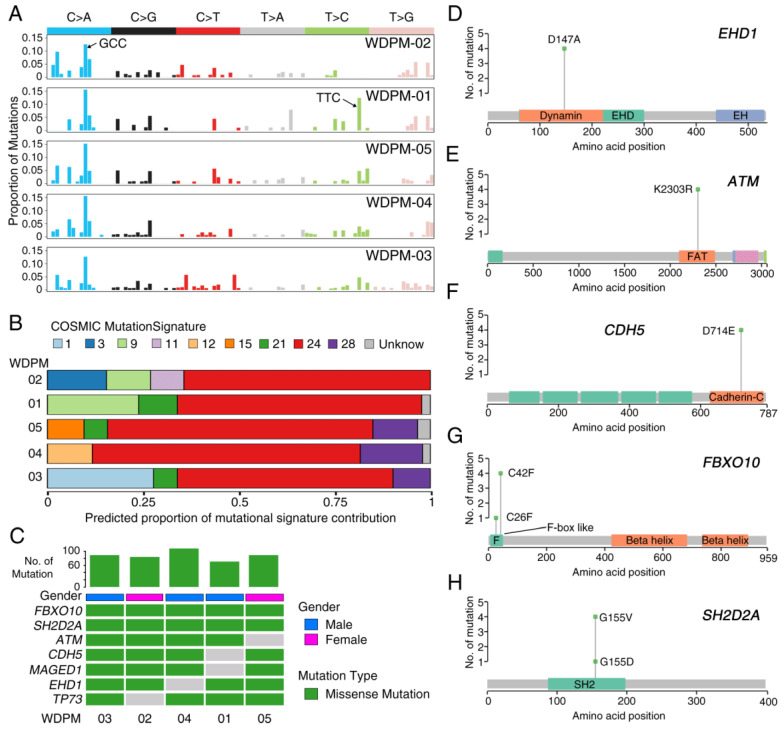
Landscape of mutations in WDPM. (**A**) Mutational signature present in WDPM. (**B**) Proportional contribution of different COSMIC mutational signature per sample. (**C**) Mutation status in WDPM. Top seven most recurrent mutations are represented in the figure. The bar plot on the top panel represents the total number of mutations detected in the respective WDPM. (**D**–**H**) Plots showing mutation distribution and the protein domains for the corresponding mutated protein.

**Figure 3 cancers-12-01568-f003:**
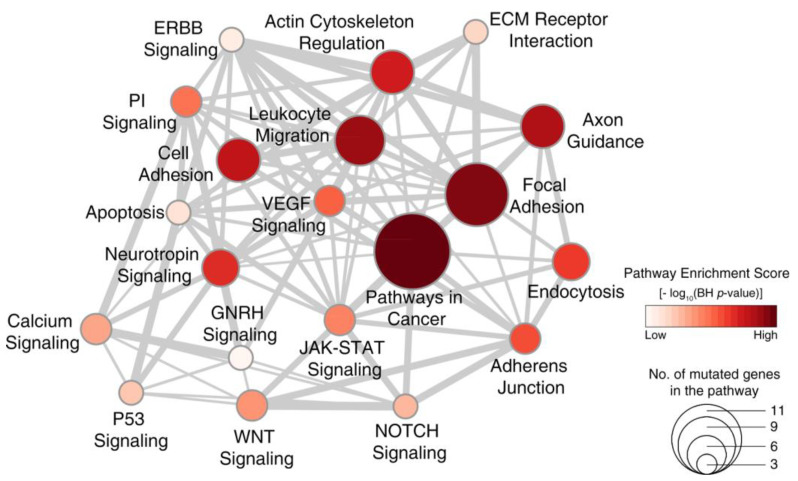
Signaling pathways dysregulated in WDPM. We performed pathway enrichment analysis using genes mutated in WDPM cases against the signaling pathways in the KEGG pathway database. The figure shows the top 20 pathways enriched with mutated genes in WDPM. Each circle represents a pathway, its size indicates the number of mutated genes targeting the pathway, and its color indicates the pathway enrichment score. The thickness of edges connecting two circles (pathways) is proportional to the number of mutated genes common between the two pathways. PI signaling: phosphatidylinositol signaling pathway.

**Figure 4 cancers-12-01568-f004:**
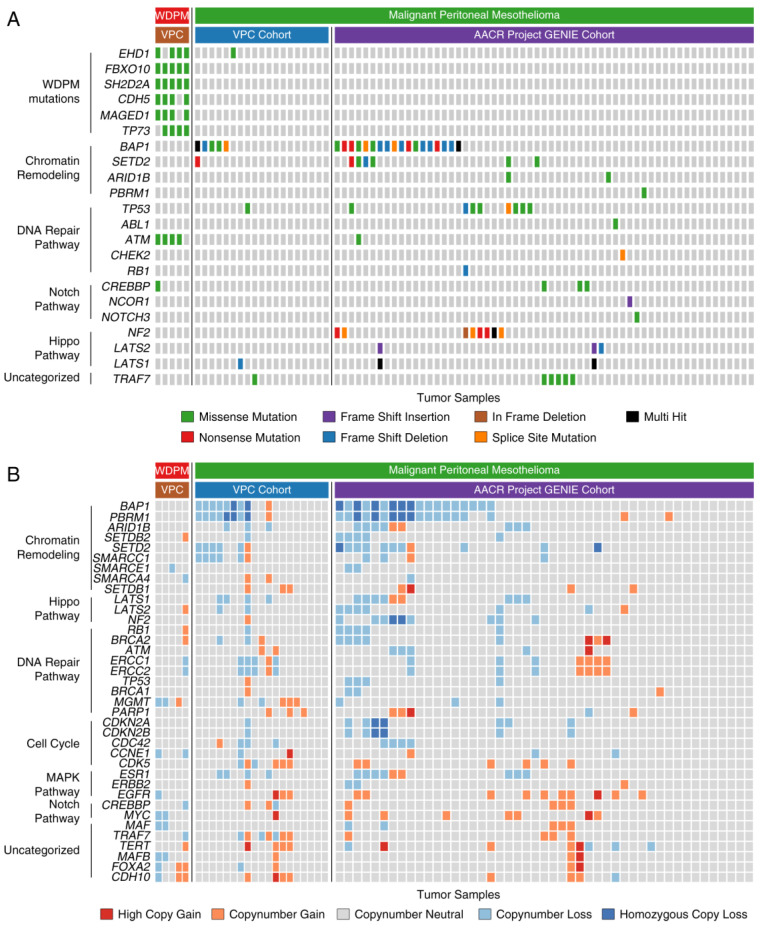
Genomic alterations in WDPM and peritoneal mesothelioma. We compared the genomic alteration profile of the WDPM cases to the peritoneal mesothelioma patient cohorts from two recently published studies, Vancouver Prostate Centre (VPC) cohort [6] and American Association for Cancer Research (AACR) project Genomics Evidence Neoplasia Information Exchange (GENIE) cohort [16]. (**A**) Oncoplot showing differences in mutation pattern between WDPM and peritoneal mesothelioma. Each column in the figure represents an individual cancer sample. (**B**) Oncoplot showing the copy number aberration status of WDPM and peritoneal mesothelioma. Each column in the figure represents an individual cancer sample.

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
