# Peer review of "Well-Differentiated Papillary Mesothelioma of the Peritoneum Is Genetically Distinct from Malignant Mesothelioma"

_cancers, 2020, doi:10.3390/cancers12061568_

Round 1
Reviewer 1 Report
I think that this paper is interesting and well written. It deals with a rare form of mesothelioma, hard to classify.
In these terms, the work is well conducted and the text is clear.
I have no observation upon methodological and experimental sections, that I think they have enough scientific soundness.
Taking into considerations the Authors' statement: "WDPM are sometimes treated with debulking cytoreductive surgery followed by hyperthermic intraperitoneal chemotherapy (HIPEC) as if they were mesotheliomas", I wondering about the possibility to add some comments in the discussion or among the conclusions dealing with the (possible) clinical use of this observations: it has sostainable costs? Could be used as a clinical exam to assess the diagnosis and relative treatment?
I think that this add coul improve the general force of this paper.
Minor note: 2.2. WDPM Mutational landscape of WDPM and 2.3. WDPM Mutational landscape of WDPM have the same heading.
Author Response
We thank the reviewer for the kind comments. We agree that a genomic approach might in future be a way of determining whether one is dealing with a malignant mesothelioma or a WDPM. However, at this point the number of cases analyzed is too small and the results too inconsistent to recommend this type of analysis at present. We have added a few lines to the end of the Discussion section to clarify this point.
The exact reasons for the discrepancy between our study and those of Stevers et al. are unclear. It is possible that the underlying populations are genetically different, particularly given the very large and diverse immigrant population in Vancouver, Canada. The analytical approach used in these two studies was also somewhat different. Stevers et al. used a targeted panel consisting of 479 cancer-related genes (UCSF500 Cancer Panel) for sequencing (Illumina HiSeq 2500), whereas we used Ion AmpliSeq™ (Thermo Fisher Scientific) Exome Sequencing which covers 18,961 genes (Ion Proton™). The overlap in the genes examined between these two studies is given in Figure S5A. Using a targeted panel provided Stevers et al. an advantage to sequence a small number of genes at a high depth (average depth = 320x, range = 33x - 722x), whereas we sequenced a large number of genes at a cost of sequencing depth (average depth = 102x). Stevers et al. reported 21 mutations covering 10 genes in 10 WDPM cases, whereas we have identified 461 mutations covering 297 genes in 5 WDPM cases. There is no overlap of the mutated genes reported in Stevers et al. and our study (Figure S5B). In fact, UCSF500 gene panel used by Stevers et al. covered only 10 mutated genes reported by our study (Figure 5C). We note that, despite high sequencing depth, no mutations in ATM (which was examined in the UCSF500 panel) were reported by Stevers et al. whereas we identified consistent ATMK2303R mutations in 4 out of 5 WDPM cases (Figure S5C). We did identify a few low confidence TRAF7 mutations, but these did not pass our mutation filtering criteria (see Table S6 and Appendix A for detail information).
These differences likely indicate genomic heterogeneity in WDPM and warrants further investigation in larger patient cohort settings. Once there are sufficient cases described with consistent results, it may be possible to use a genomic approach to decide whether an equivocal case is a WDPM or a malignant mesothelioma and to base treatment on such data.
We have corrected the headings as follows, 2.2 Mutational landscape of WDPM and 2.3. Copy number landscape of WDPM.
Reviewer 2 Report
The authors present a study of 5 cases of well differentiated papillary mesothelioma (WDPM). They have performed full exome sequencing of 18,961 genes with relatively low sequencing depth and identified recurring mutations two of which were shared by all five cases. They have also compared their findings to published sequencing data of peritoneal malignant epithelioid mesothelioma and WDPM. The mutations identified in these 5 WDPM have not been reported in malignant mesothelioma and are also different from the mutations described in the published WDPM study. Overall, the study while limited supports that WDPM is a neoplasm and genetically distinct from malignant epithelioid mesothelioma. This is important as some experts recommend aggressive treatment for WDPM that is similar to that used to treat malignant mesothelioma. Clinically most WDPM behaves in an indolent/benign fashion and identifying molecular genetic signatures that are distinct and different from malignant mesothelioma could be helpful in determining the role of adjuvant therapy. The study is well designed and illustrated and the authors have provided sufficient sequencing data in the supplementary figures and tables to support their findings.
Minor comments:
The authors may wish to add information regarding the size and anatomic location of the the 5 tumors. While WDPM is more common in women the study includes 3 cases from male patients. In our experience WDPM in men is most commonly found in the tunica vaginalis in contrast to women where WDPM is typically found in the peritoneal lining (often around the ovaries and tubes).
Author Response
We thank the reviewer for the kind comments. We have added information on the location and size of the lesions to the legend for Figure 1.
Figure 1. Histopathology of five WDPM cases used for the study. Microphotographs of histological features of WDPM stained using Haematoxylin and eosin (H&E). The panel under the dotted box represents the magnified section of the photomicrographs at ×20. The lesion sites/sizes were peritoneum, site not specified, for cases WDPM-01 (3mm), WDPM-02 (6mm), WDPM-03 (4mm), WDPM-04 mesentery (4mm), and WDPM-05 omentum (4mm).